# Muslim and Christian Women’s Perceptions of the Influence of Spirituality and Religious Beliefs on Motherhood and Child-Rearing: A Phenomenological Study

**DOI:** 10.3390/healthcare11222932

**Published:** 2023-11-09

**Authors:** Isabel del Mar Moreno-Ávila, Jose Manuel Martínez-Linares, Karim Mimun-Navarro, Carmen Pozo-Muñoz

**Affiliations:** 1Centro de Salud Zona Centro, Instituto Nacional de Gestión Sanitaria, 51003 Melilla, Spain; isamoreno89@gmail.com; 2Departamento de Enfermería, Facultad de Ciencias de la Salud, Universidad de Granada, 18071 Granada, Spain; 3Servicio Andaluz de Salud, Hospital Universitario Torrecárdenas, 04009 Almería, Spain; karimmimun@yahoo.es; 4Departamento de Psicología, Universidad de Almería, 04120 Almería, Spain; cpozo@ual.es

**Keywords:** healthcare personnel, child-rearing, religious practices, religious beliefs, relation with spirituality, lived experiences, qualitative research, phenomenology

## Abstract

(1) Background: Spirituality is a factor that plays a role in decisions related to health and illness. When a woman becomes a mother, she undergoes physical, psychological, and social changes for which healthcare professionals must provide the necessary care. However, women may feel misunderstood and stigmatized when they carry out their religious practices and express their spirituality related to motherhood. The aim of this study was to describe the experiences of women with Muslim and Christian religious ideologies on the influence of spirituality and religious beliefs in motherhood and child-rearing. (2) Methods: A descriptive phenomenological qualitative study with two groups of women of Islamic and Christian ideology, respectively. Three focus groups and in-depth interviews were conducted, recorded, transcribed, and analyzed with ATLAS.ti 7.0. An inductive analysis was carried out according to the Moustakas model. (3) Results: Three themes were identified: religious and cultural aspects that determine child-rearing, the influence of spirituality and family on the mother’s role, and the support received from healthcare personnel. (4) Conclusions: Spirituality and religious beliefs are manifested during motherhood and child-rearing in the form of infant feeding, the need for their protection, or the need for support from mothers. Healthcare personnel must be able to offer culturally competent and spiritually respectful care. Patients should not be judged based on their spirituality.

## 1. Introduction

The behavior of women who become mothers is strongly influenced by religious and spiritual beliefs during the perinatal period [1]. Religious well-being is directly and positively associated with better physical and mental health parameters, quality of life and longevity [2]. In this context, spirituality is “the individual’s perception of their position in life in the context of the culture and value system in which they live and in relation to their goals, expectations, standards, and concerns” [3]. Spirituality and religion are often used as synonyms, but they are different concepts. In contrast with religion, the term spirituality has to do with the subjective. Spirituality may be a connection with God, nature, the environment or others. On the contrary, the concept of religion is related to the traditional values and practices of a certain group of people and their beliefs [4]. Spirituality is not inherent to religion, since religion is rather a personal set or an institutionalized system of religious attitudes, beliefs, and practices [4].

Spirituality is multidimensional as a concept and is associated with inborn protection against disease and a better overall quality of life [5]. So, the application of a spiritually integrated intervention may help to enhance the mental and physical health of young and healthy nulliparous pregnant women [6]. On the other hand, religious beliefs may influence health and disease processes by promoting healthy lifestyles that avoid smoking, alcohol and other drug use [7], and they facilitate the formation of support networks [8,9], better coping with bereavement or serious illnesses [10,11] and reducing anxiety and stress levels [12,13]. However, religious beliefs may also compromise public and individual health, as in the case of some evangelical Christian groups that are against child vaccination [7]. Likewise, religious beliefs have been associated with the use of unofficial sources of information, distrust of scientific evidence, and a reluctance to vaccination during the COVID-19 pandemic [14].

Thus, it is clear that the diversity of religious beliefs entails a real challenge for health professionals, who must take care of patients but also their families, meeting their needs and adapting care to those needs [15]. Nurses and midwives are often unaware of the spirituality of the people they care for, jeopardizing holistic and person-centered care [16]. Balboni et al. [17] advocate “incorporating person-centred approaches to increase spirituality awareness among healthcare professionals as a protective factor for health, and to consider it as a research and community assessments variable, as well as in program implementation”.

For all of the above, culturally adapted attention respecting patient spirituality is crucial in the care provided by healthcare professionals. This requires working on cultural competence as a process by which health professionals achieve the ability to appropriately work within the cultural context of a family, an individual or a community, as well as the knowledge, attitudes, behaviors and even policies that enable professionals to work in different intercultural contexts [18]. Nurses should look for the most appropriate model of care for an increasingly multicultural society, and they should better understand the beliefs, practices and health problems of people from other cultures [19].

Christianity is still estimated to be the most widely practiced religion, with 2.6 billion adherents worldwide, while there are 2 billion Muslims, and forecasts put the Christian population at 3.3 billion and the Muslim population at 2.8 billion in 2050 [20].

There are many changes to be adapted for a woman who becomes a mother—physical, psychological and social changes—and healthcare professionals should provide the attention she needs [21].

Continuous monitoring of postpartum women by healthcare professionals increases women’s confidence in feeding their babies, maximizing the breastfeeding period, which may lead to the attainment of the World Health Organization recommendation of maintaining exclusive breastfeeding during the first 6 months of life [22]. However, women may feel misunderstood and stigmatized when practicing their religious beliefs and expressing their motherhood-related spirituality, as they are not understood by health personnel. Thus, “religious beliefs can be both a guide and a comfort, as well as an actual stress trigger” [23].

According to Spector’s Health Traditions Model, nurses must understand what health means in different cultures and care. In a multicultural society, this model provides a theoretical and practical framework for broadening knowledge of practices, health-related beliefs and other issues that shape the experiences of people with different cultural backgrounds [24]. Nyaloko et al. [25] state that “cultural practices are an essential part of raising children and they are important enough for healthcare providers to ensure culturally sensitive care”. As an example of this matter, Firdous et al. [26] suggest that practicing Islam has a profound impact on motherhood, influencing the timing of having children, the family size, contraceptive decision-making, fertility treatments, and other childbearing and postpartum issues. Likewise, Christianity refers to more traditional aspects of motherhood, family structure or relationships between family members [27].

Therefore, it is important to know Muslim and Christian women’s perceptions about the influence of religious beliefs on motherhood and parenting. This article answers the following research question: What are the experiences of Christian and Muslim women regarding the influence of these worldviews on motherhood and parenting?

The aim of this study is to determine how spirituality and religious beliefs influence motherhood and child rearing based on Christian and Muslim women’s experiences.

## 2. Materials and Methods

### 2.1. Study Design

A two-year descriptive phenomenological qualitative study was conducted following Husserl’s line of work. Phenomenology is a suitable methodology to describe people’s experiences regarding a phenomenon [28]. This Husserlian approach is indicated for nursing research [29,30]. The relevance of phenomenology in health sciences lies in the need to know the relationship between lived experiences and health [31]. In this case, it allowed exploring what spirituality and religious beliefs mean to mothers in parenting. The research team excluded their preconceived experiences through eidetic reduction and could describe the phenomenon from a free and unbiased point of view in a second phenomenological reduction [32]. The phenomenon in this study was the lived experience of women with two different beliefs during motherhood and parenting, specifically when spirituality is present. This descriptive phenomenological approach enabled us to describe the subjective experiences of participants and discover determining aspects of this phenomenon.

### 2.2. Participants and Background

The research was conducted at Zona Centro Health Centre, in Melilla (Spain). Melilla is a Spanish city located in North Africa, where people with Christian, Muslim and Jewish religious beliefs coexist. Approximately 46% of the population is Christian and 52% is Muslim [33]. This study was conducted from November 2019 to November 2021, and it involved two groups of participants, one of women professing Muslim religion and the other of women professing Christian religion, all of them in the postpartum period. The abovementioned health center is where pregnancy and puerperal check-ups are carried out for women—of Spanish or Moroccan origin—affiliated to the Spanish social security system, those who have a residence card, or women of other nationalities who have the right to have pregnancy, childbirth and puerperal check-ups by Organic Law 4/2000 on the rights and freedom of foreigners in Spain and their social integration. The study sample consisted of 22 Muslim women and 23 Christian women.

Recruitment was performed using purposive sampling at the third pregnancy visit (24–25 weeks of gestation) in order to avoid late abortions, and according to the following inclusion criteria: age between 18 and 45 years, residence in Melilla and having been followed up within the public health system. Any maternal or neonatal health problems that could influence parenting, any religious beliefs different from Islam or Christianity, and the expression of non-acceptance were the exclusion criteria. Non-acceptance was due to a reluctance to express personal opinions and experiences about religious beliefs and motherhood in public. There was no personal relationship with participants, except for the professional relationship developed during pregnancy monitoring.

The sociodemographic characteristics of the study subjects are included in Appendix A Table A1. As important data, Muslim women declared themselves more practicing than Christian women (100% vs. 47.1%; *p* < 0.001), they had a greater number of children (2.65 vs. 1.24; *p* < 0.001), and they had their first child at a younger age (25.82 vs. 32.12; *p* < 0.001). However, more Christian women completed the maternal training program than Muslim women (82.4% vs. 23.5%; *p* < 0.001).

### 2.3. Data Collection

Data were collected in the city of Melilla by two different methods at two different time points. First, 2 focus groups were conducted consisting of 10 women—5 Muslims and 5 Christians—purposively selected in the midwife’s office (who held a master’s degree) as it was a well-known place to the women and was where they reported feeling comfortable. Data collection was organized and agreed upon at the puerperal visit; it lasted approximately one hour for each case and served as a first approach to the study subjects and their experiences with motherhood and parenting. These sessions started with the open-ended question “Tell me everything you think was important during motherhood and the upbringing of your children”. Two researchers were present. One of them acted as an observer, who collected nonverbal communication elements in a field notebook, which was later incorporated into the data analysis.

Data from the focus groups were used to conduct in-depth interviews, which was the second data collection method. These interviews were conducted in the same place one month after the focus group disbanded, and with non-participants. For interview script elaboration, different aspects related to motherhood and parenting that arose in the focus groups were considered. Because of the research team’s daily contact with the participants during pregnancy and postpartum monitoring, conversations that took place on a daily basis helped to identify and formulate suggestions to include for clarification, if necessary. Interviews were conducted when women attended a puerperium visit and were introduced with an open-ended question focused on the research goal. Women were allowed to express themselves freely in order to create a calm and peaceful atmosphere. All interviews were audio recorded and transcribed verbatim. Transcripts were sent to participants to check for accuracy. With 14 interviews conducted, the necessary data saturation was achieved [34]. Table 1 shows the interview protocol.

### 2.4. Data Analysis

Verbatim transcripts were read by two research members at the same time as interviews were re-listened to, carried out for a better understanding of the experiences described by participants as well as to enable non-verbal communication elements from both groups of women to be included. An inductive analysis was performed according to Moustakas’ model [35], which emerged as a modification of the Stevick–Colaizzi–Keen method, commonly used in descriptive phenomenological studies. Firstly, a complete reading of the transcripts was carried out and then units of meaning were extracted in a second reading to create categories by grouping them according to the relationships established between them, thus carrying out the information coding. In the next step, categories were grouped into topics. For each topic identified in each interview, a textual or general description of the experiences described by the participants was performed. Next, a structural description of each topic was elaborated for each interview followed by a textual–structural description of each interview. Finally, all of them were integrated generating the topics identified.

Data analysis was carried out individually by two researchers, and they subsequently compared their results. Findings were returned to participants to identify their contributions and to consider the researchers’ interpretation of them. ATLAS.ti 7 Windows software (Scientific Software Development GmbH, Berlin, Germany) was used for this purpose. Table 2 shows an example of the analysis process carried out.

### 2.5. Rigor

Lincoln and Guba’s [36] quality criteria were used to assess the quality and scientific rigor of the study. Credibility was achieved through data saturation in the focus groups and interviews conducted, data triangulation by means of interviews and focus groups, research triangulation by two researchers in data analysis, and returning transcripts to participants. Transferability is supported by the abundance of details provided in this document on the context and the sociodemographic characteristics of the participants. Similarly, reliability is verified by a large number of details provided in the methodology and coherence between the design, data collection, and analysis performed. Confirmability is shown by participants’ feedback in two stages, first on the transcripts and then on the results obtained, as well as the incorporation of direct quotations from their accounts in the Results section.

### 2.6. Ethical Considerations

Ethical principles of research involving human subjects, as stated in the Declaration of Helsinki, were applied in this study. In addition, confidentiality and secrecy of personal information were respected in accordance with Organic Law 7/2021 of 26 May on data protection. Prior to conducting the interviews, a meeting was held with the Coordinator of the Women’s Care Unit at the Zona Centro Health Centre in the city of Melilla to inform her about the objectives of the study and its implementation and to obtain her authorization (date: 11 April 2019) (there is no ethics committee in Melilla). Both in the study design and in data collection procedures, the aim was always to avoid causing any harm to the participants, and it was ensured that their participation did not generate any risk. They were free to withdraw from the study without any consequences. All women signed an informed consent form and were informed about the study objectives, methodology, and the possibility of withdrawing consent at any time and without any consequences. The confidentiality of personal data was ensured by assigning an alphanumeric code to identify interviews and focus groups in accordance with Organic Law 3/2018 of 5 December on personal data protection and digital rights. Before being interviewed, participants had no doubts about the research question, the aim of the study, the data collection methods, what their participation consisted of, and the importance of their participation. Participants were free to refuse to answer questions. Only the main researcher knows the identity of the participants, and she is the only person in charge of protecting the data collected. No data appear in the interview recordings, transcripts, or this paper that could be used to identify the participants. All documents containing personal data were digitized and saved in files protected with access passwords.

## 3. Results

Three topics were identified regarding the established objective: (1) religious and cultural beliefs as determining factors in parenting, (2) the influence of family and spirituality on the mother’s role, and (3) support received from healthcare professionals.

### 3.1. Religious and Cultural Beliefs as Determining Factors in Parenting

Practicing participants reported that their religious beliefs play an important role in their lives and influence their understanding of motherhood and parenting. Both Muslim and Christian women who identified themselves as practicing reported their religious beliefs to be a great influence in their daily lives and in all aspects related to motherhood and parenting. As an example, Muslim women follow the teachings of the Quran regarding breastfeeding, which urge them to continue breastfeeding babies until they reach two years of age. All participants, but mostly Muslim ones, attributed great importance to breastfeeding as a practice during parenting, frequently mentioning it in their responses.

“I am a Muslim and it gives me so much peace and it helps me to be a better person. Of course, religion is in my every decision. It is recommended for the baby to be breastfed for at least two years”.(10D)

“I am a practicing Christian and although there is nothing as recommended breastfeeding in the Bible, there are teachings about dedication as a mother and breastfeeding decision inevitably implies dedication”.(8L)

However, the influence of spirituality is much more evident in practicing Muslim women. They choose to breastfeed because it is stated to until the child is two years of age in the Quran, and they believe everything the Quran says to be advice for being a better person. Being able to carry out its teachings entails a great reward.

“Of course, my religious beliefs are in my every decision. I still have a lot to read but I know that it is recommended for the baby to be breastfed for at least two years”.(30H)

Non-practicing participants associated the practices they performed with tradition, and they left religion outside, giving it a non-significant role. They perceived that there is confusion between tradition and religious beliefs, making it complicated to distinguish between them.

“I have been living and coexisting all my life with different cultures here. Sometimes we confuse what is done because of tradition with what is done because of religion. I am a Christian but not a practicing one”.(11A)

“I chose breastfeeding because of everything I learned in the classes during my pregnancy. I am a Christian and a practicing one in my own way, but my decision is not influenced by being one at all”.(21A)

### 3.2. Family and Spirituality Influence on Mother’s Role

For the participants, the moment for a woman to actively develop her role as a mother begins at childbirth. In addition, their perception of motherhood changed after becoming a mother for the first time. They stated that life changes drastically once the baby is born, as do their priorities. Both groups of women referred to fears about the health and well-being of their sons or daughters. They expressed that being a mother, especially in the first few days, is a tough and difficult process to handle, so they turned to their spirituality and even pray as a means to distance themselves from the negative thoughts and recover peace and a sense of protection.

“Since I became a mother, everything in me has changed. I used to be a super athlete, super vain, and I am no more. I even pray every day asking for my son to be healthy, well and protected, which I didn’t do before”.(25C)

Both Christian and Muslim women have an idealized perception about motherhood before becoming a mother for the first time, and spirituality has nothing to do with this. This idealization is the result of the messages transmitted by media and the general idea of it not being appropriate to mention the negative aspects of motherhood. However, this idealized perception changes with the arrival of new children and, although this fact makes it easier for the mother to develop her role, they also generate a greater workload. Motherhood is perceived more realistically due to previous experiences, and mothers have better knowledge about how to act when a child gets sick or when a small domestic accident occurs. This is more common in Muslim women, who usually tend to have more descendants.

“After four children, and this is the fifth, being a mother is easier, but there are more things to do at home and one forgets about oneself in the end. The more children the more work, but one gets used to it”.(24H)

There was an evident difference in the level of knowledge about postpartum and newborn care. Compared to Muslim women, Christian women attended prenatal classes or antenatal care programs to a greater extent and had a more active role in learning more about their pregnancy. For the first ones, knowledge transmitted to them by their mothers and their close presence during childbirth and postpartum provided them with the calm and security they needed.

“Going to maternal lessons and learning the basic care of the newborn has helped me a lot these days. I have applied everything I learned and read, and it has been very useful”.(29C)

“My mother has been by my side and that was so helpful for me. Since she has been a mother of six, she knows everything about this stage. Being with her has been very comforting”.(8L)

#### Family Support as a Determining Factor

The mother’s performance of their role may be facilitated by family and partner support in spirituality issues, and it can be decisive for adapting to motherhood. 

“My family is essential to me, especially my husband. I have been able to breastfeed for longer thanks to all of them”.(24H)

Both Muslim and Christian women perceived family support of the mother as something of great importance. Thanks to previous experiences of mothers and sisters who went through this same process, the emotional instability of women, especially at the beginning of parenting, improves as they feel more confident about the role they have to perform.

“He is my second son and believe me, I would not have been a mother again without them. It’s very hard and I don’t want to imagine the whole process without any help”.(21A)

Muslim women usually have a more extensive female family support network, including cousins and sisters-in-law, because Muslim families tend to be larger and meet more frequently, particularly women.

“I am the youngest of five brothers and sisters, and the truth is that my family helps me a lot, but especially the women in my family”.(7M)

Mothers-in-law play a fundamental role for Muslim women, and many of them lived together with the couple after the marriage. This coexistence becomes something normal when the mother-in-law becomes widowed and sons become their mother’s protectors, as caring for the elderly is part of Muslims’ spirituality and beliefs. Even if this is not the case, Muslim women have a close and continued relationship with their mothers-in-law at the beginning of motherhood, while this is not commonly expressed by Christian women, who meet this need with family and close friends.

“My mother-in-law also helps me and has helped me with nutrition during pregnancy; besides, I live with her. My husband is the eldest brother and my father-in-law died a few years ago, so my mother-in-law lives with us. This is very common among Muslims, taking care of the elderly. And there is a special bond between my husband and his mother, and he is very protective of her”.(20N)

“My sister and my mother are everything to me. Thanks to them and with the help of my friends, I have been able to move forward in the postpartum period. All the changes and the overall situation have been tough, but there is always family at home throughout the day and that is food for my soul”.(17M)

### 3.3. Support Received from Healthcare Professionals

When becoming a mother, healthcare professionals’ help is perceived as essential by all participants as it increases confidence both during childbirth and postpartum. Having a team of professionals who provide culturally adapted care and consider women’s spirituality is essential, and it is perceived as necessary, especially for practicing Muslim women.

“In primary care and in the hospital, everything was fine. It is true that a lactation consultant is crucial, but understanding my need to pray during childbirth is also a must. I noticed a lack of understanding about my religious beliefs and how important they are to me. Being a Muslim implies time for my faith and this entails many prayers a day, as well as other issues that transform the person. Having professionals by my side who would understand my need to continue with my religion and the need to be with God at the time of delivery would have been ideal. We need professionals who understand that people cannot be separated from their religion and that serving or caring for a person with religious beliefs implies adapting their care to these beliefs”.(14K)

Both groups expressed that counseling and knowledge about different worldviews would help to eradicate ingrained beliefs and practices with no scientific evidence. Christian women spoke about some ideas such as not holding the baby for too long because “they get used to being held,” or not co-sleeping because “babies will not want to sleep alone after that” (19M). On the other hand, Muslim women maintain habits such as giving chamomile to infants “to avoid colics” (4N) or rocking them too much “to prevent them from scratching themselves” (10D).

## 4. Discussion

The goal of this study was to describe Muslim and Christian women’s experiences regarding the influence of spirituality and religious beliefs on motherhood and child-rearing. The study revealed that there is an important role to play for religious beliefs in aspects such as breastfeeding, motherhood experience and newborn care. It also exposed the need for healthcare professionals to understand and appreciate the importance of these beliefs during this period of women’s lives.

In this study, practicing participants perceived how their religious beliefs influence some aspects of motherhood and child-rearing. Among Muslim women, pregnancy could be considered a spiritual state, whether a woman actively practices her religion or not [23]. For women with Muslim worldviews, their system of religious beliefs and values gives meaning to the choice of breastfeeding as a method of feeding their babies [37]. This relationship between religion and breastfeeding is more evident the stronger their religious convictions are [38]. The influence of religious beliefs for practicing Muslim women on the type of infant feeding chosen and its duration has been reported in other studies. The Quran’s teachings on the benefits and duration of breastfeeding [39,40] or information provided by religious leaders [40,41] were key elements for these women to choose breastfeeding. Breastfeeding is intertwined with the system of beliefs and values of Islam [37]. On the other hand, as other studies reveal, Catholic women are generally in favor of exclusive formula feeding compared to women of other religious beliefs [42].

The presence of spirituality during pregnancy, childbirth, and postpartum processes has been demonstrated in studies such as Backes et al. [43], showing that pregnancy and childbirth are experienced differently by each woman, depending on physical, mental, social, and spiritual factors. In this regard, cultural and contextual factors, as well as spirituality, are involved in the management of childbirth pain, and they give both pregnancy and childbirth a meaning [44]. According to the same study, this should be known and considered by healthcare professionals during childbirth support. 

The influence of women’s spirituality on performing the mother role was also shown in a study carried out by Prinds et al. [45], in which it is stated that motherhood creates an existential sense in women’s lives, making experiences such as death, responsibility, and vulnerability more relevant. Hence, this process is perceived as “hard”, especially at the beginning. As revealed in other studies, the perspective of a need to “make it through the first month” in primiparous women is a psychological burden and implies a series of maternal responsibilities [46], this being the time when motherhood is de-idealized [47], as highlighted in the present study. Muslim mothers, in addition, suffer higher levels of maternal stress than Jewish mothers [48] because they are not always allowed to leave the house [49]. However, Muslim women consider that mothering demands many sacrifices, which Allah will reward them for. It is a difficult process that requires effort and time [50]. During the present study, participants had an idealized idea of motherhood and child-rearing, which drastically changed after having their first baby. As stated in other studies, the postpartum period is crucial for first-time mothers, as they try to adapt to a new life [51]. A recent systematic review revealed the existence of a motherhood myth as something that mothers experienced, and it is closely related to their experience of maternal guilt. Mothers described feeling that they did not live up to their own ideals [52]. This idealization may lead mothers to consider that they must satisfy the baby’s needs, even without considering their own [53]. Moreover, it seems to be a phenomenon independent of the mother’s spirituality, since it also occurs in Japanese women. In this case, primiparous women pursued that ideal of a mother too [47].

Nevertheless, as shown in this study, for both groups of women, this idea of mothering changes after the second child, since women of any creed or ideology seem to have greater self-confidence [54]. In any case, the transition to motherhood is a highly important process that should be addressed during pregnancy monitoring, both in primiparous and multiparous women of any religious ideology [55].

Antenatal care programs influence the development of this maternal role. However, Christian women are more likely to partake in these programs while Muslim women place more value on the knowledge transmitted to them by their mothers and women in their families. This result agrees with that of other studies in which Christian women were more likely to use the services of skilled midwives in Guinea [56]; this is also a proven fact in the case of Muslims in sub-Saharan Africa [57], India [58], Bangladesh [59], Ethiopia [60], and Nigeria [61], but not those in Burundi [62].

Nevertheless, Muslim women who attended prenatal classes before giving birth for the first time stated that these classes gave them practical tools to prepare for labor and delivery, and found them helpful in providing reassurance about giving birth, relieved their fears, prepared them psychologically, and made them feel more comfortable about what the experience might entail [62]. As mentioned before, those who did not attend prenatal classes found the information they needed in consultation with family [63].

The significance of interpersonal relationships with family members during this period of women’s lives revealed in this study has also been shown in other studies. In the case of Muslim women, this relationship extends to a greater number of women in their family. This is also common among Indonesian women who had positive experiences related to the support they have received, especially from mothers, grandmothers and mothers-in-law [64], which is more obvious in the case of teenage mothers [65]. In this regard, it can be stated that spirituality positively intervenes in family relationships and in behaviors adopted during child-rearing [66,67,68]. The participants who depended on their family for support noted that this support included physical support during pregnancy, labor, and post-labor (for up to 40 days) as well as in initiating and maintaining breastfeeding [23]. Chen et al. also stated that family relationships are an essential aspect during motherhood, especially at the beginning or in transition to it, giving crucial importance to the mother-in-law role [69]. However, not every Muslim woman perceives their relationship with their mother-in-law as positive during the postpartum period—quite the opposite [70]—which is also the case among Christian women, for whom the relationship was perceived as “problematic and exhausting” [71]. This result agrees with those of the present study.

Participants from both groups perceived healthcare professionals’ help as essential as it was viewed that they create a safety net during childbirth and postpartum. To achieve this, it is important for healthcare professionals to develop cultural competence [72]. This becomes especially important in healthcare professionals who attend to maternal and child health, as meeting these needs would increase satisfaction in every woman regarding the care received [23] and would improve their relationship and feelings towards their baby, as well as promoting a better start and longer duration of breastfeeding, thus facilitating adaptation to motherhood [73]. In a study carried out in Norway, refugee women felt they were treated differently by health professionals due to their religion, and they barely understood the information provided to them [74]. In the present study, the participants did not perceive different behavior for not having the same religious beliefs as the health personnel who they were receiving care from, but they did indicate a need to be respected. Healthcare professionals who consider spirituality and being able to provide culturally adapted care will be able to address spiritual health, thanks to empathy, respect for beliefs, rituals, and symbols related to pregnancy and childbirth [26,43]. To do this, health professionals in maternity units need to attend training programs on the impact that cultural and religious practices have on the health needs of women [23,75].

In the same way, and as highlighted as a result in this study, health personnel in maternity units may eradicate false beliefs about motherhood and child-rearing. And it is the same in the case of women from other faiths [76] and migrant women [77,78].

### Limitations

This study is limited to Christian and Muslim women. Results could be different for women with other religious beliefs. This research could be complemented with the opinions and experiences of women’s partners who also raise their children, so this would be an interesting issue to study. There is no reason to believe that the testimonies provided unilaterally by women do not reflect reality since field notes collected through observation during interviews and incorporated into data analysis showed consistency between what was said and how it was said.

It is intended to continue this research focusing specifically on the methods and the importance of the development of health professionals’ cultural competence in specialties related to maternal and child health due to the lack of published studies on this subject in public health systems.

## 5. Conclusions

Spirituality is part of human nature and is present in all people throughout their entire lives. Therefore, it is also present in motherhood, a stage in a woman’s lifetime implying a great emotional burden. It is evident in the way mothers feed their babies, especially in practicing Muslim women, as well as in contemplating spirituality as a resource to ask for and to have a sense of protection against illnesses or problems that may appear.

Both Christian and Muslim women idealize motherhood before becoming mothers for the first time, something that disappears as more children are born. However, Christian women face postpartum and newborn care with further knowledge by attending antenatal care programs.

In the constantly changing period motherhood represents, receiving support from the partner, family, and mother-in-law—in the case of women of Islamic religious beliefs—eases their adaptation to postpartum life. Healthcare professionals have a meaningful place during patients’ pregnancy, childbirth and postpartum period, and it is more than necessary to care for them with a comprehensive approach during this process. Both Christian and Muslim women attribute great importance to the help and care provided by these professionals during childbirth and postpartum. Considering beliefs and spirituality during this period not only allows for good attention to their needs but also offers culturally competent and holistic care, especially for Muslim women.

As professionals who attend to women during motherhood and child-rearing, healthcare professionals must be trained to develop cultural competence that enables them to adapt the care they provide to the cultural characteristics, spirituality and religious beliefs of women in the multicultural environments in which they carry out their work. In a globalized world in which migration is increasingly frequent, health professionals in maternity units must be trained and prepared to know and understand the spirituality and religious beliefs of women related to motherhood and child-rearing.

## Figures and Tables

**Table 1 healthcare-11-02932-t001:** Interview protocol.

Interview Stage	Topic	Sample Question
Presentation	Reasons	Belief that their perspective provides knowledge that has to be globally known.
	Intentions	Conduct research with the goal of showing a real situation.
Start	Opening question	Tell me everything you think was important during motherhood and the upbringing of your children.
Development	Clarification prompts	Please tell me, in as much detail as possible, how you think your spirituality and religious beliefs have influenced the way you raise your children. How do you think motherhood is conditioned by spirituality and/or religious beliefs in a context such as the city of Melilla?
Closing	Closing question	Is there anything else you would like to tell me?
	Appreciation	Thank you very much for your collaboration. Your contributions are of great value.
	Offering	I would like to remind you that you can call me or send me an e-mail if you have any questions at all.

Source: compiled by author.

**Table 2 healthcare-11-02932-t002:** Example of the analytical process (from quote to theme).

Quote	Codes	Unit of Meaning	Sub-Theme	Theme
“Sometimes we confuse tradition and religion. I am a Christian but not a practicing one; however, I am looking forward to Easter to behold the statues”.	Religious beliefs or culture?; Muslim religious beliefs; Christian religious beliefs; practicing religion; not practicing religion; influence on breastfeeding; breastfeeding in the Quran; error in practice; ideological confusion.	Difference between culture and religious beliefs and how they intertwine, which conditions motherhood to a greater or lesser extent.		Religious and cultural beliefs as determining factors in parenting.
“In the hospital and health center, care has been good. Postpartum has been and still is a rough and difficult reality”.	Clash between expectation and reality; mother’s role; the importance of the husband; the importance of the family for Muslim women; the importance of the family for Christian women; extensive female family support; mother-in-law’s support for Muslim women; rough reality; mothering with support.	Motherhood actively begins with childbirth. It will be conditioned by the type of delivery, as well as the support she has from her partner and the number of children she has.	Family support as a determining factor	Family and spirituality influence on mother’s role.
“In primary health and in the hospital, everything has been very good. Although a lactation consultant solely dedicated to it is necessary. I have received quality care, but the result, especially in the first few days, could have been better”.	The help from health professionals; help with breastfeeding; consultants; perception of safety; adaptation to beliefs; culturally adapted care; false beliefs; respect.	The help of professionals, especially when becoming a mother, is essential to ensure a healthy and safe postpartum period. It is necessary to have updated knowledge about lactation.		Support received from healthcare professionals.

Source: compiled by author.

## Data Availability

Due to the sensitive nature of the questions asked in this study, survey respondents were assured that raw data would remain confidential and would not be shared.

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
