# Peer review of "Muslim and Christian Women’s Perceptions of the Influence of Spirituality and Religious Beliefs on Motherhood and Child-Rearing: A Phenomenological Study"

_healthcare, 2023, doi:10.3390/healthcare11222932_

Round 1

Reviewer 1 Report

Comments and Suggestions for Authors

The manuscript presented for review reports results of an empirical study looking into the mothers' subjective experiences around the relationship between religion/ spirituality and mothering. I strongly believe, that this is an important subject, this kind of research is much needed!

After careful reading of the manuscript I conclude that it is not suitable for publication in its present form. There are number of problems which require authors’ attention:

1.       The manuscript requires extensive language editing. At times the text is difficult to comprehend due to poor English (mainly errors in the way sentences are structured and mixing tenses)

2.       There are significant conceptual problems throughout the text. In the introduction the authors attempt to define religion and spirituality and make a claim that these phenomena are distinct. In my opinion the definitions provided by the authors are very general and do not constitute sufficient basis for operationalization of religion and spirituality during data collection and analysis. Later in the text religion and spirituality are used interchangeably and other terms are added to the mixture. Author speak loosely about religious practice, spiritual practice, beliefs, religious ideology  without making any attempt to define or distinguish them.

3.       Related to this is the methodological issue regarding the interview protocol. All Topics and sample questions focus on religion, there is no mention of spirituality. Is this intentional? If so, why did the authors decide to exclude “spirituality” from the protocol?

4.       In the results section there are many generalization and some contradictory statements. For example, authors report that non-practicing participants felt that their mothering practice was not influenced by religion, yet they open the next paragraph with a generalizing statement “For both Muslim and Christian women spirituality and religious ideology always  end up influencing parenting guidelines in one way or another.” Does it mean that they discovered in their data evidence that contradicts the declarations of non-practicing mothers? Or are there other reasons to make this statement? This is not clear and very confusing. The following statement: “Influence however is much more evident in practicing Muslim women.” Is also problematic. Authors support that statement by referencing breastfeeding guidelines present in the Quran and followed by Muslim women. Why is breastfeeding practice singled our like this? To make this comparative statement at least some other mothering practices should be considered? One more example of unsubstantiated and very problematic statement is: “Both Christian and Muslim women have an idealized perception about motherhood  before becoming a mother for the first time and religious ideology has nothing to do with it” Is this participants’ declaration, or authors conviction? If the latter, how would they support this? I think it would be very difficult to maintain that somehow study participants were isolated from religious discourses and images, or able to reject them completely when constructing their idealized expectations around mothering.

5.       In the discussion section the authors cite lots of literature, but do very little to connect it to their study findings. I was under impression that these references were thrown in simply because they reported findings on similar subject. There is no indication if all these studies supported authors finding, whether there were any differences and how they could be explained, or how the present study expands the existing knowledge, contributes to further understanding.

6.       Ethical considerations:

-I think it is misleading to state under the subheading “2.6. Ethical considerations” that the study was discussed and approved by the Coordinator of the Women's Care Unit at the “Zona Centro'' Health Centre. Having an approval if the institution’s manager to conduct the study on its premises has nothing to do with research ethics, it simply enables the authors to recruit participants and collect data. I understand that there is no ethics committee in the  city under study, and that this does not mean that the research was unethical, the above statement just diverts readers attention from the fact that the ethical consideration should be explained in more depth in the paper given no formal ethical approval  was obtained.

- Another issue is the declared preservation of research participants’ anonymity. I think authors field in this regard. As the GDPR guidelines state, personal data is not just names. If we have a rough estimation when the study was conducted – hence when study participants were using the services of the Women's Care Unit at the “Zona Centro'' Health Centre, their profession, religious affiliation and the age of their first birth, they are perfectly identifiable!!!

7.       The structuring of the paper makes it difficult to follow. In the introduction there is lack of balance between two target groups, for example, authors state that: “Firdous et al. [26] suggest that Muslim religion has a profound impact on motherhood—timing of having children, family size, contraceptive decision making, fertility treatments, and other childbearing  and postpartum issues” o how about Christian mothers? Also there is overemphasis on breastfeeding in the paper, whereas other mothering practices are not discussed (yet the title suggests that the topic is broader than breastfeeding). In the results section the paragraphs are very short, the sections feels very fragmented. In the discussion section, as discussed the connection between study findings and the referenced literature is not being made. In the conclusions section the answer to the main question the authors asked is drowned by other themes, such as, for example, the idealization of motherhood.

Comments on the Quality of English Language

The manuscript requires extensive language editing. At times the text is difficult to comprehend due to poor English (mainly errors in the way sentences are structured and mixing tenses)

Author Response

Manuscript ID: healthcare-2642033

Title: Perceptions of Muslim and Christian Women on the Influence of Spirituality and Religious Beliefs in Motherhood and Child Rearing: A Phenomenological Study.

Author's Reply to the Review Report (Reviewer 1)

The manuscript presented for review reports results of an empirical study looking into the mothers' subjective experiences around the relationship between religion/ spirituality and mothering. I strongly believe, that this is an important subject, this kind of research is much needed!

Thank you very much for your appreciation of our article. For us, it is very important to know that this subject is of interest for articles about it to be published.

We incorporate the responses to your suggestions in the text in green font.

After careful reading of the manuscript I conclude that it is not suitable for publication in its present form. There are number of problems which require authors’ attention:

  1. The manuscript requires extensive language editing. At times the text is difficult to comprehend due to poor English (mainly errors in the way sentences are structured and mixing tenses).

Thank you very much for your suggestion. Once the changes indicated by the reviewers have been made, a new language editing has been carried out.

  1. There are significant conceptual problems throughout the text. In the introduction the authors attempt to define religion and spirituality and make a claim that these phenomena are distinct. In my opinion the definitions provided by the authors are very general and do not constitute sufficient basis for operationalization of religion and spirituality during data collection and analysis. Later in the text religion and spirituality are used interchangeably and other terms are added to the mixture. Author speak loosely about religious practice, spiritual practice, beliefs, religious ideology without making any attempt to define or distinguish them.

Thank you very much. We have expanded and clarified the difference between spirituality and religious beliefs. Spirituality is part of the person's subjectivity, while religion includes a series of values ​​and practices that include religious beliefs. From there, in the rest of the Introduction only the terms Spirituality and Religion/Religious Beliefs are used.

  1. Related to this is the methodological issue regarding the interview protocol. All Topics and sample questions focus on religion, there is no mention of spirituality. Is this intentional? If so, why did the authors decide to exclude “spirituality” from the protocol?

That was a mistake. In the protocol that was designed, the questions referred to spirituality and religious beliefs. That is why they have been modified as they appear in Table 2.

  1. In the results section there are many generalization and some contradictory statements. For example, authors report that non-practicing participants felt that their mothering practice was not influenced by religion, yet they open the next paragraph with a generalizing statement “For both Muslim and Christian women spirituality and religious ideology always end up influencing parenting guidelines in one way or another.” Does it mean that they discovered in their data evidence that contradicts the declarations of non-practicing mothers? Or are there other reasons to make this statement? This is not clear and very confusing. The following statement: “Influence however is much more evident in practicing Muslim women.” Is also problematic. Authors support that statement by referencing breastfeeding guidelines present in the Quran and followed by Muslim women. Why is breastfeeding practice singled our like this? To make this comparative statement at least some other mothering practices should be considered? One more example of unsubstantiated and very problematic statement is: “Both Christian and Muslim women have an idealized perception about motherhood  before becoming a mother for the first time and religious ideology has nothing to do with it” Is this participants’ declaration, or authors conviction? If the latter, how would they support this? I think it would be very difficult to maintain that somehow study participants were isolated from religious discourses and images, or able to reject them completely when constructing their idealized expectations around mothering.

Thank you very much for your suggestion. We have re-ordered the content and quotations in theme "3.1. Religious and cultural determinants of parenting". We think that in this way, the content follows a more logical order and does not create confusion, distinguishing between the perceptions of practicing and non-practicing women.

We have used breastfeeding as an example of parenting practice because of the importance that the participants gave to it. It appeared in his speeches repeatedly. This is why it has been highlighted in the content of the theme.

Regarding the idealization of motherhood, it was the participants who referred to this. To clarify, we have added a sentence explaining why this idealized perception is due.

  1. In the discussion section the authors cite lots of literature, but do very little to connect it to their study findings. I was under impression that these references were thrown in simply because they reported findings on similar subject. There is no indication if all these studies supported authors finding, whether there were any differences and how they could be explained, or how the present study expands the existing knowledge, contributes to further understanding.

Thank you very much for your considerations. The discussion has been thoroughly reviewed to ensure that the results have been discussed correctly and that all aspects discussed are consistent with the results obtained.

  1. Ethical considerations:

-I think it is misleading to state under the subheading “2.6. Ethical considerations” that the study was discussed and approved by the Coordinator of the Women's Care Unit at the “Zona Centro'' Health Centre. Having an approval if the institution’s manager to conduct the study on its premises has nothing to do with research ethics, it simply enables the authors to recruit participants and collect data. I understand that there is no ethics committee in the  city under study, and that this does not mean that the research was unethical, the above statement just diverts readers attention from the fact that the ethical consideration should be explained in more depth in the paper given no formal ethical approval  was obtained.

In order to improve this subsection, we have expanded and provided more data on the ethical considerations that were followed throughout the research process.

- Another issue is the declared preservation of research participants’ anonymity. I think authors field in this regard. As the GDPR guidelines state, personal data is not just names. If we have a rough estimation when the study was conducted – hence when study participants were using the services of the Women's Care Unit at the “Zona Centro'' Health Centre, their profession, religious affiliation and the age of their first birth, they are perfectly identifiable!!!

To ensure that the participants cannot be identified, the columns for profession, city of origin and type of breastfeeding have been eliminated in the table of sociodemographic characteristics. These data are of low relevance to the study, but eliminate the possibilities for identifying the participants.

  1. The structuring of the paper makes it difficult to follow. In the introduction there is lack of balance between two target groups, for example, authors state that: “Firdous et al. [26] suggest that Muslim religion has a profound impact on motherhood—timing of having children, family size, contraceptive decision making, fertility treatments, and other childbearing and postpartum issues” o how about Christian mothers? Also there is overemphasis on breastfeeding in the paper, whereas other mothering practices are not discussed (yet the title suggests that the topic is broader than breastfeeding). In the results section the paragraphs are very short, the sections feels very fragmented. In the discussion section, as discussed the connection between study findings and the referenced literature is not being made. In the conclusions section the answer to the main question the authors asked is drowned by other themes, such as, for example, the idealization of motherhood.

Thank you very much your considerations. We have re-structured the paper:

- We have compensated with another bibliographical reference on Christianity.

- We have placed more emphasis on breastfeeding because it was an aspect of motherhood and parenting that women highlighted in their speeches. However, mention is also made of the false beliefs that women have about other aspects of motherhood and parenting.

- The discussion has been thoroughly reviewed to ensure that the results have been discussed correctly and that all aspects discussed are consistent with the results obtained.

- In the conclusions section, all the results of the study are briefly mentioned and the importance of the cultural competence of health professionals in maternity units is highlighted.

Comments on the Quality of English Language

The manuscript requires extensive language editing. At times the text is difficult to comprehend due to poor English (mainly errors in the way sentences are structured and mixing tenses).

Thank you very much for your suggestion. Once the changes indicated by the reviewers have been made, a new language editing has been carried out.

Reviewer 2 Report

Comments and Suggestions for Authors

Overall, you are to be applauded for your hard work. The research is well done and relevant. I suggest that the authors reread the article for needed edits to clean up some of the writing. Otherwise, well done!

Comments on the Quality of English Language

Overall, the article is well written. However, it would help if the authors were to read through it again and make edits for clarity. I will give a handful of examples below that need to be rewritten:  

Page 1, line 27: The final sentence of the abstract is vague and should be rewritten. "Spirituality must not be judged by them."

Page 1, line 40-41: "...religion is more likely a personal set or institutionalized system of religious attitudes..." use some commas or reword

Page 1, line 43: "...Spirituality "in" nature is multidimensional..." add "in"

These are some examples that need clarity and the whole paper should be examined for similar issues.

Author Response

Manuscript ID: healthcare-2642033

Title: Perceptions of Muslim and Christian Women on the Influence of Spirituality and Religious Beliefs in Motherhood and Child Rearing: A Phenomenological Study.

Author's Reply to the Review Report (Reviewer 2)

Overall, the article is well written. However, it would help if the authors were to read through it again and make edits for clarity.

Thank you very much for your appreciation of our article. We incorporate the responses to your suggestions in the text in red font.

I will give a handful of examples below that need to be rewritten: 

Page 1, line 27: The final sentence of the abstract is vague and should be rewritten. "Spirituality must not be judged by them."

We have modified this sentence: “Spirituality should not be judged by them.”

Page 1, line 40-41: "...religion is more likely a personal set or institutionalized system of religious attitudes..." use some commas or reword

Thank you very much. We have reworded this sentence.

Page 1, line 43: "...Spirituality "in" nature is multidimensional..." add "in"

Added “in” in this sentence: “Spirituality in nature is multidimensional and is associated with inherent protection against disease and better overall quality of life [5].”

These are some examples that need clarity and the whole paper should be examined for similar issues.

We have carried out a new grammar and editing review of the language. We hope it is correct.

Reviewer 3 Report

Comments and Suggestions for Authors

This study addresses a very important issue in women's health and subjectivities, and uses a very interesting set of data. However, its data analysis is very shallow and provides little substantial thematic codes. It lacks a coding frame that such a content analytical study would naturally produce, except for the three very sketchy codes. I recommend that it could develop more substantive thematic codes to be considered for publication.

Comments on the Quality of English Language

n.a.

Author Response

Manuscript ID: healthcare-2642033

Title: Perceptions of Muslim and Christian Women on the Influence of Spirituality and Religious Beliefs in Motherhood and Child Rearing: A Phenomenological Study.

Author's Reply to the Review Report (Reviewer 3)

This study addresses a very important issue in women's health and subjectivities, and uses a very interesting set of data.

Thank you very much for your appreciation of our article. We incorporate the responses to your suggestions in the text in blue font.

However, its data analysis is very shallow and provides little substantial thematic codes. It lacks a coding frame that such a content analytical study would naturally produce, except for the three very sketchy codes. I recommend that it could develop more substantive thematic codes to be considered for publication.

Thank you very much for this consideration. We have expanded the number of codes used in the qualitative analysis and which are provided in Table 3.
